# Transcriptomic and Metabolomic Analysis of a Fusidic Acid-Selected *fusA* Mutant of *Staphylococcus aureus*

**DOI:** 10.3390/antibiotics11081051

**Published:** 2022-08-03

**Authors:** Sushim K. Gupta, Richard F. Pfeltz, Brian J. Wilkinson, John E. Gustafson

**Affiliations:** 1Department of Biochemistry and Molecular Biology, Oklahoma State University, Stillwater, OK 74078, USA; sushim.gupta@okstate.edu; 2BD Life Sciences, Microbiology Research and Development, Sparks, MD 21152, USA; richard.pfeltz@bd.com; 3School of Biological Sciences, Illinois State University, Normal, IL 61761, USA; bjwilkin@ilstu.edu; 4Department of Biology, New Mexico State University, Las Cruces, NM 88003, USA

**Keywords:** *Staphylococcus aureus*, fusidic acid-resistance, respiration, metabolomics, transcriptional profiling

## Abstract

Physiological experimentation, transcriptomics, and metabolomics were engaged to compare a fusidic acid-resistant *Staphylococcus aureus* mutant SH10001st-2 to its parent strain SH1000. SH10001st-2 harbored a mutation (H457Y) in the gene *fusA* which encodes the fusidic acid target, elongation factor G, as well as mutations in a putative phage gene of unknown function. SH10001st-2 grew slower than SH1000 at three temperatures and had reduced coagulase activity, two indicators of the fitness penalty reported for *fusA*-mediated fusidic acid- resistance in the absence of compensatory mutations. Despite the difference in growth rates, the levels of O_2_ consumption and CO_2_ production were comparable. Transcriptomic profiling revealed 326 genes were upregulated and 287 were downregulated in SH10001st-2 compared to SH1000. Cell envelope and transport and binding protein genes were the predominant functional categories of both upregulated and downregulated genes in SH10001st-2. Genes of virulence regulators, notably the *agr* and *kdp* systems, were highly upregulated as were genes encoding capsule production. Contrary to what is expected of mid-exponential phase cells, genes encoding secreted virulence factors were generally upregulated while those for adhesion-associated virulence factors were downregulated in SH10001st-2. Metabolomic analysis showed an overall increase in metabolite pools in SH10001st-2 compared to SH1000, mostly for amino acids and sugars. Slowed growth and metabolite accumulation may be byproducts of *fusA* mutation-mediated protein synthesis impairment, but the overall results indicate that SH10001st-2 is compensating for the H457Y fitness penalty by repurposing its virulence machinery, in conjunction with increasing metabolite uptake capacity, in order to increase nutrient acquisition.

## 1. Introduction

Methicillin-resistant *Staphylococcus aureus* (MRSA) causes a large percentage of hospital-acquired infections and community-acquired infections in healthy individuals with no prior exposure to healthcare [1]. In 2017, *S. aureus* caused 119,247 bloodstream infections and 19,832 deaths in the United States [2]. The first MRSA emerged in 1961, shortly after the introduction of methicillin for clinical use, and over subsequent decades, MRSA strains have developed resistance to virtually all β-lactams and other drug classes. Because of the evolution of MRSA, vancomycin became the therapeutic choice to treat infections caused by MRSA, and for some multidrug- resistant MRSA strains, vancomycin represented a treatment of last resort. With the eventual emergence of vancomycin-intermediate *S. aureus* (VISA) and vancomycin-resistant *S. aureus* (VRSA), the use of vancomycin for the treatment of MRSA infections has been curtailed [3].

Fusidic acid (FA) is considered to be an important component of treatment options for infections caused by MRSA, VISA, or VRSA, in part due to a relatively low prevalence of FA resistance in *S. aureus* [4]. FA is a steroid antibiotic that is used topically and systemically in combination therapy for chronic staphylococcal skin or bone and joint infections, respectively [4]. The target of FA is elongation factor G (EF-G), an essential translation factor required for peptide translocation and then ribosome recycling during protein synthesis. The FA mechanism of action leads to stalled protein synthesis by locking EF-G to the ribosome, preventing its turnover [5]. FA is approved for use in Europe, the United Kingdom, Australia, New Zealand, Canada, and a number of Asian countries, but not in the United States.

In clinical environments, *S. aureus* FA resistance arises via chromosomal mutations in *fusA*, the gene encoding EF-G, or via the acquisition of horizontally-transferred elements such as the FA resistance gene 1 (*far1* or *fusB*) [6] and *far1* homologues (*fusC* and *fusD*). Mutations in *fusA* result in structural changes in EF- G that decrease the affinity between FA and its target [7,8]. Inducible *far1* and its homologues can be chromosomal or plasmid-borne and are all thought to impart FA resistance by supporting a target-protection mechanism whereby FA is blocked from binding to EF-G [9]. Additional mechanisms, such as mutations within *rplF* that encodes ribosomal protein L6 [10] and efflux pump activity [11] have also been shown to contribute to FA resistance in the laboratory. Staphylococci isolates with FA MICs ≤ 1 mg/L are considered susceptible and isolates with MICs > 1 mg/L are considered resistant [12].

Previously, transcriptional profiling was used to characterize the FA stimulon of pan-susceptible *S. aureus* strain SH1000. This fusidic acid stimulon shared strong similarities with the cold shock and stringent stress responses, led to the upregulation of the accessory gene regulator (*agr*) virulence operon and elements of the *walKR* cell wall metabolism regulon, and led to the altered expression of a large number of genes encoding protein synthesis and degradation functions [13].

In the present study, a SH1000 FA-resistant mutant (SH10001st-2) was compared to SH1000 at the genomic, transcriptomic, metabolomic, and phenotypic levels, following growth in drug-free liquid media. The data produced provides additional information on novel physiology underlying the *fusA* mutation- mediated FA resistance mechanism in an important pathogen.

## 2. Results and Discussion

### 2.1. FA MICs and Comparative Genomic Sequencing (CGS)

CGS revealed a single nucleotide polymorphism (SNP) in the *fusA* gene in FA-resistant mutant SH10001st-2 (FA MIC = 32 mg/L) when compared to parent strain SH1000 (FA MIC = 0.125 mg/L) (Table 1). This SNP led to a common EF-G H457Y amino acid substitution that is associated with clinically acquired FA resistance which can reach FA MICs as high as 64 mg/L [14]. Therefore, both the *fusA* mutation and FA MIC of SH10001st-2 are representative of those described in the literature for clinical FA-resistant strains. CGS revealed 4 additional SNPs selected in SH10001st-2 that resulted in three amino acid substitutions in a gene encoding a DUF1381 superfamily protein (Table 1). DUF1381 superfamily proteins are a group of *S. aureus* bacteriophage proteins with no known function.

### 2.2. Phenotypic Characterization of SH10001st-2

Growth curves of SH10001st-2 revealed that this mutant grew slower than parent SH1000 at three temperatures (25 °C, 37 °C, and 42 °C) (Figure 1). However, respiration measurements revealed that SH1000 produced a similar amount of CO_2_ (178 ± 24.8 uL CO_2_/h/OD unit) and consumed a similar amount of O_2_ (159.2 ± 16.4 uL O_2_/h/OD unit) as SH10001st-2 (166.2 ± 19.74 uL CO_2_/h/OD unit; 154.6 ± 24.47 uL O_2_/h/OD unit; *n* = 6, *p* > 0.05).

SH1000 demonstrated no hemolytic activity on blood agar, whereas SH10001st-2 produced zones of hemolysis 7.8 ± 0.4 mm in diameter (*n* = 3). Suspensions of SH10001st-2 and plasma required 195 ± 0 min for plasma clots to appear versus only 75 ± 0 min with SH1000 (*n* = 3, *p* < 0.05). The fitness costs represented by the reduced growth rates and coagulase activity observed in SH10001st-2 have previously been reported to result from fusidic acid resistance-mediating amino acid substitutions in FusA [15]. Compensatory *fusA* intragenic mutations that correct these fitness costs have also been reported [15,16]. Based on the reduced growth rate and coagulase activity observed in SH10001st-2, it is not likely that the four SNPs in the DUF1381 superfamily gene represent fitness-compensating mutations.

### 2.3. Overview of the SH10001st-2 Transcriptome

The 613 total gene expression differences between parent SH1000 and SH10001st-2 are listed in Appendix A. RT-qPCR was utilized to validate the upregulation and downregulation of select genes whose expression was altered and detected by transcriptional profiling (Table 2).

A summary of transcriptomic results by 16 gene product functional categories is presented in Table 3. Of the genes upregulated in SH10001st-2, 45.1% were represented by three functional categories: cell envelope, regulatory functions, and transport and binding proteins. Among genes downregulated in SH10001st-2, 36.3% of genes were from four categories: cell envelope, transport and binding proteins, biosynthesis of cofactors, prosthetic groups and carriers, and DNA metabolism. On both the upregulated and downregulated lists, the cell envelope and transport and binding proteins categories included the greatest number of characterized gene products, together accounting for 36.2% of upregulated and 27.5% of downregulated gene sets (Table 3). Genes encoding hypothetical uncharacterized proteins collectively represented 37.7% of upregulated and 36.9% of downregulated genes in SH10001st-2.

Protein synthesis genes accounted for 3.1% of upregulated and 3.1% of downregulated genes, and the associated transcription and protein fate categories accounted for 5.2% of both the upregulated and downregulated genes (Table 3). Interestingly, the protein synthesis gene *prfC* was upregulated in SH10001st-2 by a factor of 2.26 (Appendix A). This gene encodes peptide chain release factor 3, which shares significant sequence homology with EF-G, and is involved with the fidelity of protein synthesis [17,18].

### 2.4. Highly Upregulated and Downregulated Genes in SH10001st-2

Table 4 lists all genes upregulated by ≥10-fold in SH10001st-2 compared to parent SH1000. Virulence factors and virulence regulatory elements dominated the 26 upregulated genes list (Table 4), including all 8 genes in the cell envelope functional category: seven encoding capsular polysaccharide synthesis enzymes plus the delta-hemolysin encoded by RNAIII of the *agr* global quorum sensing/regulatory system that is directly involved with virulence gene expression by *S. aureus* [19,20]. The upregulated virulence genes also included 2 lipase genes and two genes (SACOL0212 and SACOL214) encoding gene products involved with fatty acid oxidation. In addition, 4 regulatory virulence genes encoded by RNAII of the *agrBCDA* operon and 2 regulatory genes encoding the *kdpDE* two-component regulatory system were also on this list. More than 150 genes have been identified as regulated by the *agr* system [20]. Finally, three of the four genes in the transport and binding proteins category encoded components of the KdpFABC system, and one of the four hypothetical protein category gene products was a putative phenol-soluble modulin secreted virulence factor (Table 4). KdpDE in *S. aureus* acts as a transcriptional regulator of virulence factors by means of a mechanism that possibly involves sensing external potassium levels via KdpFABC; the Kdp system itself is upregulated by the *agr* system [21,22].

Table 5 lists the 11 genes downregulated by ≥10-fold in SH10001st-2 versus parent SH1000. These consisted of 3 genes encoding cell envelope-associated virulence factors (*isaB*, SACOL0089 and *sspB*) as well as 3 genes associated with nitrogen metabolism (*narK*, *nirR,* and *narG*). Virulence regulators were notably absent from the list of highly downregulated genes.

### 2.5. SH10001st-2 Expression of Virulence Factors

Virulence-associated genes differentially expressed in SH10001st-2 compared to parent strain SH1000 are found in Appendix A. Upregulated genes for secreted proteins, 10 of which encode degradative enzymes (e.g. proteases, hemolysins, or lipases), outnumbered the downregulated genes by more than 3:1 (13 upregulated versus 4 downregulated). All 11 genes encoding surface adhesion proteins were downregulated in SH10001st-2 (Appendix A). A majority of the surface proteins not involved in adhesion have immune evasion functions. More than twice as many genes encoding non-adhesion surface proteins were upregulated as downregulated (ratio 19:8) in SH10001st-2, almost entirely represented by the upregulation of all 16 genes of the *cap* operon (*capA-capP*). The *cap* operon, as well as many secreted enzymes, are upregulated by the *agr* system [20] and KdpDE also regulates capsular polysaccharide production [23]. Expression of genes encoding several global transcriptional regulators of virulence from the SarA family [24] were differentially regulated (e.g., *sarR*, *sarS*, and *rot*) (Appendix A), notably the repressor of toxins (*rot*) which was downregulated. *rot* plays a role in the regulation of the *kdp* system by *agr* [21]. It has previously been reported that the *agr* operon was upregulated by FA challenge and that the *agr* operon [13] and the virulence regulatory gene *sarA* are required for the full expression of intrinsic low-level FA resistance [25]. The increased hemolysis and reduced coagulase activity exhibited by SH10001st-2 can be explained by the upregulation of three hemolysin-encoding genes (*hla*, *hlb*, and *hld*) and the downregulation of the gene encoding staphylocoagulase (*coa*) in SH10001st-2 (Appendix A).

Iron acquisition is a key virulence strategy employed to counter iron sequestration, a common host defense against pathogens [26]. The most highly downregulated gene in SH10001st-2 encodes an iron compound transporter, SmpB (Table 5), and there were 9 iron/heme transport-associated genes that were upregulated (e.g., *isdC*, *isdG*, *srtB*, and heme permeases) (Appendix A and Table 5). A gene encoding oleate hydratase, which is a virulence factor that promotes immune evasion by modifying host fatty acids provided by secreted lipases [27], was also downregulated (Table 5). We note that these gene expression patterns were inconsistent with a strictly upregulated virulence response interpretation.

### 2.6. Metabolomics of SH1000 vs SH10001st-2

Fifty-three characterized metabolites were significantly altered in SH10001st-2 when compared to parent strain SH1000 (*p* < 0.05). Table 6 lists the 42 metabolites present at greater concentrations in SH10001st-2, 20 of which demonstrated ≥2-fold increases. Table 7 lists the 12 metabolites present at lower concentrations in SH10001st-2, 4 of which demonstrated ≥2-fold decreases. The 17 amino acids and 13 sugars in Table 6 together accounted for more than 60% of the metabolites that increased in concentration in SH10001st-2. Five metabolites had ≥10-fold increased concentrations in SH10001st-2: glucosamine and four amino acids, including *N*-acetyl-serine with a 70-fold increase and serine with a striking 221-fold increase. Nine metabolites were not detected in SH1000 but accumulated in SH10001st-2, including three amino acids (asparagine, homocysteine, and threonine) and five sugars that included glucose-6-P, fructose, ribitol, and ribose (Table 6). Only two metabolites were detected in SH1000 but not in SH10001st-2; arginine and linoleic acid (Table 7).

A number of alterations in the expression of genes encoding transport and binding proteins likely contributed, or partially contributed, to the increase in amino acids and sugars in SH10001st-2. While 11 genes encoding amino acid permeases and uptake systems (*aapA*, *brnQ3*, *gltT*, SACOL1367, SACOL1392, SAR1419, SAS2274, and SAV1380) and oligopeptide transporters (SAV0727, SAV1380, and SAV1381) were upregulated in SH10001st-2, 8 other genes in this category (SACOL1476, SACOL2453, SAR2503, SAV0722, SAS0283, SAV2412, *opuCA*, and *proP*) were downregulated (Appendix A). At the same time, there were more amino acid biosynthesis genes downregulated (*argF*, *asd*, *cysE*, *glyA*, *hisF*, SAS0418, and SAS2563) than upregulated (SACOL2044 and SAV1737) in SH10001st-2 (Appendix A). With regard to carbohydrate transporters, 3 (SACOL2146, SACOL2552, and *treP*) were downregulated and another 2 (SAV0192 and *ptaA*) were upregulated (Appendix A).

The downregulation of SH10001st-2 genes directly involved in pyruvate metabolism (e.g., *ddh*, encoding D-lactate dehydrogenase, and *adh1*, encoding alcohol dehydrogenase) (Appendix A and Table 5) may explain the elevated levels of pyruvic acid in SH10001st-2 (Table 6). The amino sugar glucosamine is a key building block for capsular polysaccharide and cell wall biosynthesis that is associated with central intermediary carbohydrate metabolism [28]. The 11-fold concentration increase in glucosamine in SH10001st-2 may be required to support the anabolic demands associated with the upregulated expression of genes encoding capsular biosynthetic enzymes.

The serine family of amino acids are prominent among those with increased concentrations in SH10001st-2. Downregulation of *cysE* (Appendix A), which participates in the conversion of serine to cysteine, would contribute to the accumulation of free serine, as would upregulation of *aapA* (Appendix A), which encodes a D-serine/D-alanine/glycine transporter. However, downregulation of *glyA* (Appendix A), which catalyzes the production of serine from glycine, would contribute to the observed accumulation of glycine but at the expense of the serine pool. Furthermore, pyruvate can be synthesized directly from serine [29]. In Gram- negative organisms, *N*-acetyl-serine, which is produced from the cysteine precursor *O*-acetyl-serine whose own synthesis is catalyzed by CysE, serves a regulatory function with respect to sulfur assimilation and metabolism [30]. Unfortunately, the relationship between amino acid metabolism and metabolite pools is not well understood in staphylococci, although the regulation of amino acid and carbohydrate metabolism involves the *agr* system’s response regulator AgrA [31].

## 3. Materials and Methods

### 3.1. Culture Conditions, Growth Curves and Antibiotic Susceptibility

Bacteria were propagated in Luria–Bertani broth (LB) (Difco, Detroit, MI, USA) with shaking (200 rpm, 37 °C) or on LB agar (LBA), unless otherwise stated. Permanent stock cultures were in LB containing 20% glycerol at −80 °C; working stock LBA cultures were stored at 4 °C. Overnight cultures were initiated from single colonies and grown at 37 °C with shaking (200 rpm) overnight. To initiate growth curves, broth cultures were inoculated with an overnight culture to reach an initial OD580nm of 0.04 which was monitored over time with triplicate cultures (25 °C, 37 °C, or 42 °C; 200 rpms). Unless otherwise noted, chemicals were obtained from Sigma Alrich Co., St. Louis, MO, USA.

Agar dilution MICs for parent and mutant strains were determined on Mueller-Hinton agar (MHA). Strains were grown overnight at 37 °C in Mueller–Hinton broth (MHB) with shaking (200 rpm). The resulting suspensions were adjusted to an OD_625nm_ of 0.01 and 2 L aliquots were plated onto MHA containing a two-fold FA concentration series from 0.0625–512 mg/L. MICs were then determined after 24 hr of incubation at 37 °C according to CLSI guidelines.

FA resistance was laboratory-selected in *S. aureus* strain SH1000 of the NCTC 8325 lineage [32]. SH1000 mutant colonies appeared in 2 mg/L FA at a mutation frequency of 2.2 × 10^−9^ after 24 hr, and a FA-resistant mutant colony (SH10001st-2) was selected and passed through drug-free media multiple times before any experiments were performed.

### 3.2. Metabolic Activity

Comparison of metabolic activity between SH1000 and SH10001st-2 was performed by taking O_2_ and CO_2_ measurements, as previously described [33]. Briefly, *S. aureus* were grown in LB at 37 °C with shaking (200 rpm) and O_2_ and CO_2_ measurements were determined in triplicate by distributing 2.5 mL of early exponential-phase (OD_580nm_= 0.2) cultures separately into glass boats within 50 mL airtight glass analytical chambers sealed with brass plugs. The chambers were flushed for 1 min at a flow rate of 150 mL/min with CO_2_-free air of known O_2_ concentration and left sealed for 20 min. The OD_580nm_ was then determined at the end of the experiment for each of the triplicate samples.

### 3.3. Hemolysis and Coagulase Test

Overnight 5 mL MHB cultures were diluted with MHB to an OD580nm to 0.1 and serially diluted by factors of 1 × 10^7^ and 1 × 10^8^ with sterile MHB. Trypticase soy agar with 5% sheep blood plates (BBL, Sparks, MD, USA) were inoculated with 0.1 mL of each dilution and incubated at 37 °C for 24 hr. Hemolytic activity was assessed by measuring the diameters of the zones of hemolysis in triplicate in isolated colonies on the plate.

Triplicate 5 mL LB overnight cultures were diluted with LB to an OD580nm of 1.0, then 50 L of each standardized culture was transferred to sterile glass tubes containing 0.5 ml of rehydrated coagulase plasma and gently mixed. The tubes were then incubated at 37 °C and periodically examined by gently tipping the tube. Coagulase activity was assessed by determining the clotting amount and time in triplicate.

### 3.4. DNA and RNA Purification and cDNA Synthesis

*S. aureus* chromosomal DNA was extracted using the spooling method [34], and RNA for quantitative real-time PCR (qRT-PCR) and microarray analysis was isolated from mid-exponential phase cultures (OD580nm = 0.7) using a bead mill homogenization procedure [35] following pretreatment of cell pellets with RNA Protect (Qiagen Inc., Germantown MD, USA). cDNAs were synthesized from DNA-*free* (Ambion, Austin, TX, USA) treated RNA using Moloney murine leukemia virus Super Script III reverse transcriptase (Invitrogen), as previously described [33].

### 3.5. Comparative Genomic Sequencing

Comparative-genome sequencing (CGS) was performed with the complete genomes of *S. aureus* SH1000 and SH10001st-2 using a tiling microarray-based service provided by NimbleGen Systems Inc. (Madison, WI, USA) that can identify up to 95% of all single nucleotide polymorphisms (SNPs) and insertion-deletions, followed by sequencing of the genomic alterations identified (Table 1). A complete description of the genomic tiling microarray design and the CGS comparison methodology used has already been described [36].

### 3.6. DNA Microarray and Quantitative Real-Time PCR Analyses (qRT-PCR)

For microarray analysis, cDNA samples prepared from SH1000 and SH10001st-2 were labeled with Cy3 or Cy5 post-labeling reactive dye following the manufacturer’s suggestions (Amersham Biosciences, Piscataway, NJ, USA). Microarray experiments were performed in duplicate, and fluorophore dyes were swapped to produce dual cDNA samples to minimize dye bias for each strain cDNA preparation analyzed. *S. aureus* DNA microarrays version 4 produced by the Pathogen Functional Genomics Resource Center (https://www.ncbi.nlm.nih.gov/geo/query/acc.cgi?acc=GPL7072) were used for hybridization and array analysis. Hybridized arrays were scanned with a GenePix 4000B Microarray Scanner (Axon Instruments, Union City, CA, US) and array TIFF images were analyzed as previously described [33]. Gene upregulation or downregulation of ≥ 2.0-fold was considered significant. The microarray data have been deposited in NCBI’s Gene Expression Omnibus and is accessible through GEO Series accession number GSE12210 (https://www.ncbi.nlm.nih.gov/geo/query/acc.cgi?acc=GSE12210).

The iCycler iQ Real-Time PCR Detection System (Bio-Rad Laboratories, Hercules, CA, USA) and iQ SYBR Green Supermix (Bio-Rad) were utilized for qRT-PCR of control and test cDNAs. The expression level of each sample in triplicate was normalized using 16S rDNA as an internal control and expression ratios were determined using the 2^-∆∆Ct^ method, as previously described [33]. All primers utilized for qRT-PCR are described in Appendix A.

### 3.7. Metabolite Extraction and Analysis

Metabolites were extracted from SH1000 and SH10001st-2 cultures in triplicate, as previously described [37]. Briefly, overnight cultures were used to inoculate 125 mL volumes of LB to reach an initial OD580nm of 0.01 which were then incubated with shaking (37 °C, 200 rpm) until an OD580nm of 0.7 was reached. Cells from each sample were then harvested using centrifugation (10,000× *g*, 5 min, 4 °C) and washed once in PBS (137 mM NaCl, 2.7 mM KCl, 10 mM Na2HPO4, 2 mM KH2PO4, pH = 7.4). Metabolic quenching was achieved by adding 0.5 mL of cold methanol (−20 °C) before storing samples at −80 °C. Metabolite analysis was performed by the Roy J. Carver Biotechnology Center, University of Illinois at Urbana-Champaign, Urbana, IL, USA, as previously described [37]. Metabolite relative concentrations were normalized using 100 mg of dry cell weight, and statistically significant alterations in metabolite levels were determined and reported as the average ± standard error of the mean for all three replicates.

## 4. Conclusions

The *fusA* mutation in SH10001st-2 resulted in relatively few protein metabolism gene expression differences, even though *fusA* mutations can lead to reduced protein biosynthesis [38]. SH10001st-2 demonstrated a slowed growth rate and reduced coagulase activity, indicating a fitness penalty reported to occur in *fusA* mutants without compensatory mutations [15,16]. Clearly, the four SNPs in SACOL0358 did not fully compensate for the fitness cost that resulted from the *fusA* mutation in SH10001st-2. 

During exponential growth the *S. aureus* virulence regulatory system upregulates the production of surface proteins and capsular polysaccharides, which are primary immune-evasive virulence factors, to support tissue colonization. In post exponential growth or stationary phase, *S. aureus* then shifts to produce secreted enzymes and toxins required to extract nutrients and facilitate spread into nearby tissues [39].

Overall, our results indicated that mid-exponential SH10001st-2 cultures appeared to behave as if the cultures were in a post-exponential or stationary phase state, since secreted exoprotein genes were upregulated while, except for the *cap* operon, surface virulence protein genes were downregulated. It is likely that the virulence gene expression alterations observed were mediated by the increased expression of genes encoding the *agr-kdpDE* regulatory network in SH10001st-2. Increased capsular gene expression and capsule biosynthesis would be supported by the increased sugar accumulation in SH10001st-2. Virulence associated exoprotein genes can encode hemolysins, proteases, and lipases which degrade cells and macromolecules in order to provide metabolites which are then accumulated by *S. aureus* to support further growth [39,40,41]. The upregulation of virulence associated exoprotein genes could therefore have influenced the increased metabolite pools in SH10001st-2. The accumulation of amino acids in SH10001st-2 would have also been influenced by the *fusA* mutation and subsequent reduction in protein synthesis. Based on the slow growth of SH10001st-2, we assumed the CO_2_ production and O_2_ consumption would be greater in SH1000 compared to SH10001st-2, however respirometric data revealed no significant difference. This suggested that the energy produced by respiration in SH10001st-2 could be redirected from cellular growth to support hampered protein synthesis impacted by the mutated FusA and/or to support the increase in virulence gene expression. The redirected energy could also energize uptake systems resulting from the altered cell envelope and transport and binding proteins gene expressions which would also support the increased metabolite pools observed in SH10001st-2.

## Figures and Tables

**Figure 1 antibiotics-11-01051-f001:**
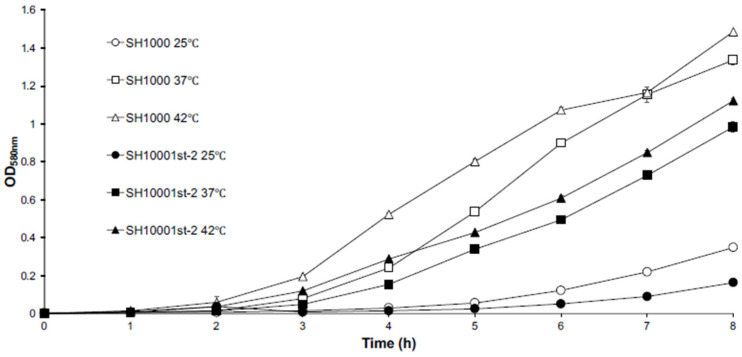
Growth curves of SH1000 (open symbols) and SH10001st-2 (closed symbols) at 25 °C (
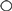
, 
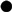
) 37 °C (
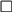
, 
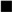
), or 42 °C (
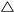
, 
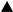
). All data shown are the mean of triplicate experiments and error bars represent the standard deviation.

**Table 1 antibiotics-11-01051-t001:** SNPs identified by comparative genome sequencing in SH10001st-2.

Gene	Protein Encoded	Locus ID	SNP *	Amino Acid Change
*fusA*	Elongation factor G	SACOL0593	C^617,228^ → T^617,228^	H^457^ → Y^457^
	DUF1381 superfamily	SACOL0358	A^371,671^ → T^371,671^	N^36^ → I^36^
			T^317,672^ → A^371,672^	N^36^ → I^36^
			G^371,676^ → A^371,676^	E^37^ → K^37^
			A^371,685^ → C^371,685^	K^40^ → Q^40^

***** Nucleotide positions are based on strain COL genome.

**Table 2 antibiotics-11-01051-t002:** qRT-PCR validation of select SH10001st-2 genes identified as altered via microarray.

		Fold-Change in Gene Expression	
Gene	Locus	Microarray	RT-PCR
*cap5E*	SACOL0140	10.4	2.1
*adh1*	SACOL0660	−7.0	−4.3
*crtN*	SACOL2576	−2.4	−2.4
*ddh*	SACOL2535	−12.2	−23.6
*spa*	SACOL0095	−8.9	−1448.0

**Table 3 antibiotics-11-01051-t003:** Functional category gene expression differences in SH10001st-2 compared to SH1000.

Functional Category	Upregulated Genes			Downregulated Genes		
	Number of Genes	% of Genes	>5% of Total	Number of Genes	% of Genes	>5% of Total
Amino acid biosynthesis	2	0.6%		8	2.8%	
Biosynthesis of cofactors, prosthetic groups, and carriers	5	1.5%		17	5.9%	5.9%
Cell envelope	66	20.2%	20.2%	46	16.0%	16.0%
Cellular processes	3	0.9%		9	3.1%	
Central intermediary metabolism	3	0.9%		13	4.5%	
DNA metabolism	9	2.8%		17	5.9%	5.9%
Fatty acid and phospholipid metabolism	8	2.5%		2	0.7%	
Mobile and extrachromosomal element functions	2	0.6%		1	0.3%	
Protein fate	7	2.1%		7	2.4%	
Protein synthesis	10	3.1%		9	3.1%	
Purines, pyrimidines, nucleosides, and nucleotides	7	2.1%		8	2.8%	
Regulatory functions	29	8.9%	8.9%	9	3.1%	
Signal transduction	0	0.0%		0	0.0%	
Transcription	0	0.0%		2	0.7%	
Transport and binding proteins	52	16.0%	16.0%	33	11.5%	11.5%
Hypothetical proteins/Unknown function/Unclassified	123	37.7%	37.7%	106	36.9%	36.9%
**Totals**	326	100.0%	82.8%	287	100.0%	76.2%

**Table 4 antibiotics-11-01051-t004:** Genes upregulated ≥ 10-fold in SH10001st-2 compared to SH1000.

Locus ID	Gene	Protein Encoded	Fold Increase
**Cell envelope**			
SACOL0136	*cap5A*	capsular polysaccharide synthesis Cap5A	18.31
SACOL0137	*cap5B*	capsular polysaccharide synthesis Cap5B	13.74
SACOL0138	*cap5C*	capsular polysaccharide synthesis Cap5C	21.30
SACOL0140	*cap5E*	capsular polysaccharide synthesis Cap5E	10.40
SACOL0146	*cap5K*	capsular polysaccharide synthesis Cap5K	14.59
SACOL0147	*cap5L*	capsular polysaccharide synthesis Cap5L	10.73
SACOL2022	*hld*	delta-hemolysin precursor	12.60
**Fatty acid and phospholipid metabolism**			
SACOL0390	NA	lipase-2 precursor, interruption-C	11.32
SACOL2694	*gehA*	triacylglycerol extracellular lipase-1 precursor	19.27
SACOL0212	NA	putative 3-hydroxyacyl-CoA dehydrogenase	11.99
SACOL0214	NA	putative long-chain-fatty-acid-acetyl-CoA ligase	10.51
**Regulatory functions**			
SACOL2026	*agrA*	accessory gene regulator A	20.27
SACOL2023	*agrB*	accessory gene regulator B	17.15
SACOL2024	*agrD*	accessory gene regulator D	14.35
SACOL2025	*argC2*	accessory gene regulator C	22.63
SACOL1032	NA	competence transcription factor ComK	13.76
SACOL2070	*kdpD*	two-component system sensor histidine kinase KdpD	10.50
SACOL2071	*kdpE*	two-component system response regulator KdpE	11.24
**Transport and binding proteins**			
SACOL2068	*kdpA*	potassium-transporting ATPase subunit A	26.07
SACOL2066	*kdpC*	potassium-transporting ATPase, C subunit	16.28
SACOL2069	*kdpF*	potassium-transporting ATPase, F subunit	29.64
SACOL1993	NA	putative ABC-2 type transport system permease	12.24
**Hypothetical proteins/unknown function/unclassified**			
SACOL1187	NA	phenol-soluble modulin beta antibacterial protein	10.36
SACOL0492	NA	hypothetical protein	18.67
SACOL0493	NA	hypothetical protein	12.20
SACOL2065	NA	hypothetical protein	13.81

**Table 5 antibiotics-11-01051-t005:** Genes downregulated ≥ 10-fold in SH10001st-2 compared to SH1000.

Locus ID	Gene	Protein Encoded	Fold Decrease
**Cell envelope**			
SACOL2660	*isaB*	immunodominant surface antigen B	−12.24
SACOL0089	NA	oleate hydratase (putative myosin-cross-reactive antigen)	−12.27
SACOL1056	*sspB*	cysteine protease precursor staphopain B	−10.80
**Central intermediary metabolism**			
SACOL2535	*ddh*	D-lactate dehydrogenase	−12.16
SACOL2395	*narG*	respiratory nitrate reductase, alpha subunit	−13.51
**Regulatory functions**			
SACOL2399	*nirR*	nitrite reductase transcriptional regulator NirR	−10.86
**Transport and binding proteins**			
SACOL1144	*smpB*	probable transmembrane protein SmpB iron compound ABC transporter	−23.21
SACOL0310	NA	nucleoside permease NupC, putative	−11.35
SACOL1476	NA	basic amino acid/polyamine antiporter, APA family	−10.63
SACOL2525	NA	lantibiotic ABC transporter ATP-binding protein	−12.61
SACOL2386	*narK*	nitrite extrusion protein	−14.55

**Table 6 antibiotics-11-01051-t006:** Metabolites increased (*p* ≤ 0.05) in SH10001st-2 compared to SH1000.

Metabolite Class	Metabolite	Metabolite Relative Concentration/Gram Dry Weight (Mean ± SE)		Fold Increase SH10001st-2/SH1000
		SH1000	SH10001st-2	
Amines & polyamines	glucosamine	4.5 ± 0.3	49.1 ± 2.4	10.91
	N-acetylglucosamine	12.3 ± 0.9	39.4 ± 2.6	3.20
	tyramine	4.6 ± 0.2	8.9 ± 0.2	1.93
	5-methylthioadenosine	11.4 ± 1.9	20.0 ± 2.3	1.75
Amino acids	asparagine	ND *	15.7 ± 1.7	
	aspartic acid	3488.5 ± 215.9	9848.3 ± 298.4	2.82
	cysteine	2.2 ± 0.2	6.8 ± 0.8	3.09
	glutamine	31.5 ± 2.3	408.1 ± 19.2	12.95
	glycine	116.4 ± 10.8	1252.9 ± 244.9	10.76
	homocysteine	ND	1.9 ± 0.1	
	homoserine	6.3 ± 0.3	22.3 ± 1.4	3.53
	isoleucine	374.8 ± 38.4	617.3 ± 69.0	1.64
	leucine	475.5 ± 28.9	800.6 ± 61.6	1.68
	N-acetyl-serine	12.2 ± 1.5	858.3 ± 73.8	70.35
	phenylalanine	182.7 ± 12.8	301.0 ± 21.9	1.64
	proline	8544.3 ± 579.9	19,899.2 ± 1871.3	2.32
	proline-like	139.9 ± 19.4	262.3 ± 14.6	1.87
	serine	11.5 ± 0.7	2,551.2 ± 196.7	221.84
	threonine	ND	101.4 ± 4.2	
	tryptophan	1.0 ± 0.2	5.7 ± 0.3	5.7
	valine	560.7 ± 34.5	803.5 ± 72.1	1.43
Polar organic acids	aminomalonic acid	ND	8.4 ± 0.4	
	citric acid	7.9 ± 1.2	39.9 ± 3.1	5.05
	fumaric acid	7.2 ± 0.7	42.6 ± 8.4	5.91
	malic acid	19.6 ± 0.7	32.6 ± 4.1	1.66
	phosphonic acid	18.9 ± 1.5	58.9 ± 1.2	3.11
	phosphoric acid	18,144.9 ± 570.4	23,007.2 ± 1620.9	1.26
	pyruvic acid	2.9 ± 0.3	7.0 ± 0.4	2.41
Sugars	1-methyl-beta-D-galactopyranoside	3.4 ± 0.4	16.6 ± 1.4	4.88
	2(1H)-Pyrimidinone, 1-ribofuranosyl-5-P	ND	15.7 ± 1.5	
	fructose	ND	3.7 ± 0.6	
	glucose	1.4	3.4 ± 0.5	2.42
	glucose-6-P	ND	3.4 ± 0.4	
	glycerol	474.9 ± 46.9	2223.0 ± 108.1	4.68
	glycerol-2-P	4.1 ± 1.0	7.3 ± 0.4	1.78
	glycerol-3-P	185.7 ± 6.1	248.6 ± 10.3	1.33
	ribitol	ND	2.9 ± 0.1	
	ribose	ND	5.0 ± 0.2	
	ribose-5-P	2.1 ± 0.1	5.3 ± 0.3	2.52
	sorbitol	8.1 ± 0.4	51.5 ± 5.8	6.35
	sorbitol-6-P	12.6 ± 2.1	23.6 ± 2.3	1.87

* ND: not detected.

**Table 7 antibiotics-11-01051-t007:** Metabolites decreased (*p* ≤ 0.05) in SH10001st-2 compared to SH1000.

Metabolite Class	Metabolite	Metabolite Relative Concentration/Gram Dry Weight (Mean ± SE)		Fold Decrease SH1000/ SH10001st-2
		SH1000	SH10001st-2	
Amines & Polyamines	adenosine	50.8 ± 2.0	15.3 ± 2.1	−3.32
	galactosamine	25.6 ± 4.2	14.9 ± 3.6	−1.71
	hydroxycarbamic acid	9.6 ± 0.4	6.8 ± 1.0	−1.41
Amino acids	pyroglutamic acid	2291.2 ± 68.0	1631.5 ± 103.1	−1.40
	arginine	3.7 ± 0.4	ND *	
Polar organic acids	2-hydroxyglutaric acid	67.5 ± 11.6	13.0 ± 0.5	−5.19
	2-hydroxyphosphinyl	2.1 ± 0.1	1.1 ± 0.1	−1.91
	gluconic acid	24.0 ± 1.6	9.0 ± 0.8	−2.66
	glyceric acid	4.6 ± 0.4	2.8 ± 0.2	−1.64
	linoleic acid	2.1 ± 0.1	ND	
	malonic acid	5.3 ± 0.4	1.8 ± 0.2	−2.94
	succinic acid	166.0 ± 13.6	23.2 ± 2.8	−7.15

* ND: not detected.

## Data Availability

The microarray data can be found at: https://www.ncbi.nlm.nih.gov/geo/query/acc.cgi?acc=GSE12210.

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
