# Peer review of "Transcriptomic and Metabolomic Analysis of a Fusidic Acid-Selected fusA Mutant of Staphylococcus aureus"

_antibiotics, 2022, doi:10.3390/antibiotics11081051_

Round 1
Reviewer 1 Report
Dear authors,
I reviewed your manuscript which compared the differences in phenotypic, gene expression and metabolomics changes between a fusidic acid resistant mutant SH10001st-2 and its parent strain SH1000 of Staphylococcus aureus. Comparative Genomic Sequencing (CGS) revealed a point mutation in FusA and additional 4 mutations in an unknown function protein, DUF1381. Transcriptome analysis revealed several up- and down-regulated genes of different cellular process in SH10001st-2 compared to its parent strain and select genes were further validated by qRT-PCR. Metabolomics profiling correlated with differences in growth profiles of above strains. Additionally, cell based assays hemolysis, coagulase test and metabolic activity by measuring O2/CO2 level was monitored.
The findings are interesting and this study will further enhance better understanding of resistance mechanism to fusidic acid in S. aureus.
There are a few shortcomings though and I feel that these minors should be addressed to improve the submitted manuscript.
Have you tried to supply a wild copy of fusA gene in trans by plasmid to restore the observed growth and other phenotype claimed to be associated with only FusA (H457Y)? Line 127-129
Title- fusA mutant or a mutant of Staphylococcus aureus resistant to fusidic acid is more appropriate?
Figure 1: are these value statistically significant? Whether this is data set of one experiment or an average of biological/technical replicates? Fig 1- Legend for 42 oC is missing.
Table 2: What could be a reason for such drastic differences in expression of spa in qRT-PCR compared to microarray? How many replicates were performed for qRT-PCR is not clear (SD is missing).
Line 110-114: Given the reason slow growth yet similar metabolic activity, it might be worth to asses if O2 consumption rate is higher in mutant than wild type?
Line 364: OD580= 0.07 or 0.7?
Line 319-320: Unit displayed are in µL?
Author Response
Reviewer 1.
- Reviewers comment: The findings are interesting and this study will further enhance better understanding of resistance mechanism to fusidic acid in aureus.
Authors response: We thank the reviewer for this comment.
- Reviewers comment: Have you tried to supply a wild copy of fusA gene in trans by plasmid to restore the observed growth and other phenotype claimed to be associated with only FusA (H457Y)? Line 127-129
Authors response: We agree with the reviewer and intend to follow up on this, but funding for the project has dried up at this moment. We intend to complete this experiment in the future with additional phylogenetically distinct mutants.
- Reviewers comment: Title- fusA mutant or a mutant of Staphylococcus aureus resistant to fusidic acid is more appropriate?
Authors response: We agree with the reviewer and have retitled the manuscript: Transcriptomic and Metabolomic Analysis of a Fusidic Acid-Selected fusA Mutant of Staphylococcus aureus.
- Reviewers comment: Figure 1: are these value statistically significant? Whether this is data set of one experiment or an average of biological/technical replicates? Fig 1- Legend for 42 oC is missing.
Authors response: All these growth curves were completed in triplicate and the error bars (which you cannot always see since the data is so tight) represent the standard deviation. To highlight this, we have placed; “All data shown is the mean of triplicate experiments and error bars represent standard deviation” – in the figure legend. We think that some of the legend was corrupted when it was transferred to the MDPI document the 42oC legend was removed/altered – which will be noted with the resubmission.
- Reviewers comment: Table 2: What could be a reason for such drastic differences in expression of spa in qRT-PCR compared to microarray? How many replicates were performed for qRT-PCR is not clear (SD is missing).
Authors response: Because we used the 2-∆∆Ct or Livak method (See https://www.sciencedirect.com/science/article/pii/S1046202301912629?via%3Dihub), we did not apply statistical analyses. These experiments were however completed with triplicate samples, and we have performed a great deal of these analyses over the years. Yes, the downregulation of spa data was higher with qRT-PCR compared to the microarray and we have seen these discrepancies in previously investigations using the same techniques, however the direction of gene expression “up or down” always correlates. It is also known that the two techniques can demonstrate variation, and it could be the platforms or biological sampling that we used are the reason the data looks different. We add: “It is well documented that both qPCR and microarray analysis have inherent pitfalls, that may significantly influence the data obtained from each method.” (see https://www.ncbi.nlm.nih.gov/pmc/articles/PMC1779618/)
- Reviewers comment: Line 110-114: Given the reason slow growth yet similar metabolic activity, it might be worth to asses if O2 consumption rate is higher in mutant than wild type?
Authors response: This was assessed in the manuscript: “Respiration measurements however revealed that SH1000 produced a similar amount of CO2 (178 ± 24.8 ul CO2/h/OD unit) and consumed a similar amount of O2 (159.2 ± 16.4 ul O2/h/OD unit) as SH10001st-2 (166.2 ± 19.74 ul CO2/h/OD unit; 154.6 ± 24.47 ul O2/h/OD unit; n = 6, p > 0.05).”
- Reviewers comment: Line 364: OD580= 0.07 or 0.7?
Authors response: We thank the reviewer for their oversight, it is 0.7 and this is corrected in the revised manuscript.
- Reviewers comment: Line 319-320: Unit displayed are in µL?
Authors response: We thank the reviewer for their oversight, it is ul and this is corrected in the revised manuscript. This alteration occurred as well when transferred to the MDPI format.
Reviewer 2 Report
Review of Gupta et al. Transcriptomic and Metabolomic Analysis of a fusA Mutant of Staphylococcus aureus.
The MS is a catalog of the transcriptional and metabolomic changes observed in a fusidic acid resistant mutant of Staph. aureus. As such it is of some interest to those working in the S. aureus and antibiotic resistance fields. A huge number of genes were altered and it's unclear what all of these transcriptional changes mean, other than serving as a salutary reminder that seemingly modest genetic changes can have profound effects on gene expression. What is lacking are some clear statements identifying the impetus for this work, what the expectations or predictions were at the outset and whether any of these were satisfied by the accumulated data.
Some small points.
1. Some doubling times (calculated from Figure 1) would be helpful
2. Lines 167-171. RF3 is NOT involved in stop codon recognition. That's the function of RF1 and RF2. RF3 recycles RF1 and 2 off the post-termination ribosome, thereby increasing their efficiency and it also seems to be involved in tRNA dropoff phenomena.
3. The authors seem to be rather careless in the choice of some references, at least from the ones I've checked.
(i) Ref 36 doesn't seem to describe the agar dilution method for determining MICs (lines 294-8).
(ii) Line 166-8:
"Since the mutated EF- G imparting FA resistance leads to slowed protein synthesis [20-22], it was surprising that the genes in these functional categories did not represent the highest number of genes 168 affected."
There are several problems here. Reference 20 is a paper from Diarmaid Hughes's lab, describing compensatory mutations in Salmonella EF-G. It does not describe the H457Y mutation, nor does it describe protein synthesis rates of any EF-G mutants. Find the proper references to support your claim, or remove this statement. References 21-22 are about ribosomal protein S12 mutants, which are not relevant to the topic under discussion.
Finally, I don't follow the logic of why EF-G alterations should provoke changes in protein synthesis components, unless you're invoking a demand-accumulation model, in which case you should state this.
4. Re-write the (horrible) sentence " The European Committee...... " (lines 70-71) so that it's intelligible.
Author Response
Reviewer 2. Comments and author responses.
- Reviewer comment: The MS is a catalog of the transcriptional and metabolomic changes observed in a fusidic acid resistant mutant of aureus. As such it is of some interest to those working in the S. aureus and antibiotic resistance fields.
Authors response: We thank the reviewer for this comment.
- Reviewer comment: What is lacking are some clear statements identifying the impetus for this work, what the expectations or predictions were at the outset and whether any of these were satisfied by the accumulated data.
Authors response: The impetus of the work was to determine how the transcriptome and metabolome are affected by mutations selected for by fusidic acid, and therefore somewhat of a fishing trip. Our expectations were: the mutations would impart a fitness cost and reduced respiratory activity, that genes associated with protein synthesis would be altered (since the mutated EF-G does alter protein synthesis), and while we did not fully know what to expect from the metabolomics data, we did expect to see changes in amino acid concentration, which we did. What was surprising was that we found large alterations in genes encoding virulence, cell envelope and transport, and binding proteins in the fusA mutant. Since we observed the agr and kdp system upregulation, we then hypothesized that the mid-exponential phase mutant culture appeared more similar to a stationary phase culture. This stationary phase state in turn led to an increase in metabolite accumulation – in part due to the upregulation of genes encoding secreted virulence degradative enzymes in conjunction with the altered expression of genes encoding cell envelope and transport, and binding proteins. Because of these findings, we found we could not simply ascribe the large increase in amino acids in the fusA mutant to a slowdown in protein biosynthesis, which was unexpected.
- Reviewers comment: Some doubling times (calculated from Figure 1) would be helpful.
Authors Response: We thank the reviewer for their comment yet we do not believe doubling time data would improve our presentation. We feel the growth curves themselves demonstrate that SH10001st-2 does not grow as well as SH1000 at 3 temperatures. Each point in Fig. 1. shows the mean of triplicate experiments and the bars represent the standard deviation. This has been clarified in the Figure 1. legend.
- Reviewers comment: Lines 167-171. RF3 is NOT involved in stop codon recognition. That's the function of RF1 and RF2. RF3 recycles RF1 and 2 off the post-termination ribosome, thereby increasing their efficiency and it also seems to be involved in tRNA dropoff phenomena.
Authors response: We thank the reviewer for their comment and have corrected our previous sentence with: This gene encodes peptide chain release factor 3, which shares significant sequence homology with EF-G, and is involved with the fidelity of protein synthesis [23,24]. We also added another recent reference that clarifies the function of release factor 3.
- Reviewers comment: The authors seem to be rather careless in the choice of some references, at least from the ones I've checked. (i) Ref 36 doesn't seem to describe the agar dilution method for determining MICs (lines 294-8).
Authors response: We thank the reviewer for this oversight and have replaced the reference with “according to CLSI guidelines”.
- Reviewers comment: (ii) Line 166-8: "Since the mutated EF- G imparting FA resistance leads to slowed protein synthesis [20-22], it was surprising that the genes in these functional categories did not represent the highest number of genes 168 affected." There are several problems here. Reference 20 is a paper from Diarmaid Hughes's lab, describing compensatory mutations in Salmonella EF-G. It does not describe the H457Y mutation, nor does it describe protein synthesis rates of any EF-G mutants. Find the proper references to support your claim, or remove this statement. References 21-22 are about ribosomal protein S12 mutants, which are not relevant to the topic under discussion. Finally, I don't follow the logic of why EF-G alterations should provoke changes in protein synthesis components, unless you're invoking a demand-accumulation model, in which case you should state this.
Authors response: We thank the reviewer again for their oversight; the references were indeed incorrect. We have also removed this sentence since it does not add to the overall findings.
- Reviewers comment: Re-write the (horrible) sentence " The European Committee...... " (lines 70-71) so that it's intelligible.
Authors response: We agree with the reviewer and have changed the sentence to read: Staphyloccoci isolates with FA MICs of ≤ 1 mg/L are considered susceptible and isolates with MICs > 1 mg/L are considered resistant [13].